# Destabilisation of the Subpolar North Atlantic prior to the Little Ice Age

Beatriz Arellano-Nava [1], Paul R. Halloran [1] ✉, Chris A. Boulton [1], James Scourse[2], Paul G. Butler [2], David J. Reynolds [2] & Timothy M. Lenton [1]

The cooling transition into the Little Ice Age was the last notable shift in the climate system prior to anthropogenic global warming. It is hypothesised that sea-ice to ocean feedbacks sustained an initial cooling into the Little Ice Age by weakening the subpolar gyre circulation; a system that has been proposed to exhibit bistability. Empirical evidence for bistability within this transition has however been lacking. Using statistical indicators of resilience in three annually-resolved bivalve proxy records from the North Icelandic shelf, we show that the subpolar North Atlantic climate system destabilised during two episodes prior to the Little Ice Age. This loss of resilience indicates reduced attraction to one stable state, and a system vulnerable to an abrupt transition. The two episodes preceded wider subpolar North Atlantic change, consistent with subpolar gyre destabilisation and the approach of a tipping point, potentially heralding the transition to Little Ice Age conditions.

The examination of natural changes in the Earth's climate system during the last millennium can improve our understanding of the mechanisms behind natural climate variability and its response to external events. The transition from the warm Medieval Climate Anomaly (MCA) to the Little Ice Age (LIA) was the last notable shift in the climate system before anthropogenic forcing became the significant factor shaping climate[1]. Global environmental reconstructions suggest that LIA cooling was most prominent in the Northern Hemisphere[1] and occurred first in the Arctic and Northern Europe during the 13th century[2]. It has been hypothesised from model experiments that sea ice to ocean feedbacks in the Arctic-North Atlantic region played an essential role in amplifying an initial transient cooling, potentially triggered by volcanic activity[3–6], leading to a reconfiguration of the system and the persistence of a cooler state[3–8].

Studies agree that an initial export of Arctic sea-ice into key convective regions of the North Atlantic weakened ocean convection. Reduced convection may have weakened the Atlantic Meridional Overturning Circulation (AMOC)[4–7] and/or the subpolar gyre (SPG) circulation strength[4,7,8], leading to a reduction in poleward heat transport, ultimately reinforcing sea-ice expansion through a positive feedback[3–8]. Both elements, the AMOC and SPG, have been shown to exhibit bistability in numerical[9–12] and theoretical models[13,14], with

transitions stimulated by changes in convection. Currently, both systems are considered at risk of being tipped into a weak state by anthropogenic climate change[12,15]. Although some environmental reconstructions suggest that the North Atlantic circulation system weakened during the MCA-LIA transition[16–18], it is unclear whether it exhibited tipping point behaviour.

The action of positive feedbacks can weaken the negative feedbacks that maintain a system in a stable attractor[19]. In such a scenario, the system becomes more sensitive to perturbations and takes longer to recover from them, meaning its resilience decreases[19,20]. Slowness in recovery manifests as reduced rates of change and increased persistence of extreme events[21]. Thus, a statistical approach to detecting changes in resilience consists of measuring trends in temporal autocorrelation and variance[21,22]. A combined increase in both metrics indicates that the system is losing resilience[20] due to the weakening of negative feedbacks[19], and may indicate the approaching of a tipping point[21]. This method requires long-term, regularly spaced, high-resolution input data[20], a set of attributes that few observational or paleoenvironmental reconstructions of the marine environment possess.

Here, we use three previously published annually resolved marine proxy records that span the last millennium, to obtain insights into the

[1]Global Systems Institute and Geography Department, University of Exeter, Exeter, UK. [2]College of Life and Environmental Sciences, University of Exeter, Penryn, Cornwall, UK. ✉e-mail: P.Halloran@exeter.ac.uk

changes in resilience in the Arctic-North Atlantic through the MCA-LIA transition. These three proxy records are derived from increment width[23] and stable isotopic composition ($\delta^{18}O$ and $\delta^{13}C$)[24,25] measurements on annual growth bands from shells of the long-lived bivalve *Arctica islandica*, collected from the North Icelandic Shelf. This location, situated close to the Polar Front where relatively warm north-flowing waters meet colder and fresher Arctic waters, is highly sensitive to changes in the North Atlantic circulation[24] (Fig. 1). Our results reveal the existence of two episodes of decreasing resilience prior to the Little Ice Age. Both episodes are aligned with wider change across the subpolar North Atlantic, likely linked to the destabilisation of the subpolar gyre.

## Results and discussion

### Changes in resilience throughout the last millennium

Environmental factors determine the properties of the *Arctica islandica* shell increments, such as width and aragonite isotopic composition[26], analogous to the records stored within tree rings. Thus, each proxy record encodes year-to-year fluctuations in a particular set of features of the marine environment. Oxygen isotope records ($\delta^{18}O_{shell}$; Fig. 2a) track the temperature and $\delta^{18}O$ of ambient seawater[24], and at regional scales, due to the salinity-like behaviour[27] of $\delta^{18}O$, the relative influence of different water masses[28]. Carbon isotope series ($\delta^{13}C_{shell}$; Fig. 2b) encode changes in the $\delta^{13}C$ composition of dissolved inorganic carbon in seawater, integrating signals from drivers such as primary productivity and remineralisation rates[25,26], with a second-order contribution from ontogenetic and species-specific effects[25]. Shell growth records (Fig. 2c) integrate the variability of all factors that influence bivalve growth, including age, temperature, and pelagic food supply[23]. Increment widths are represented with unitless growth indices that result from processing to remove age-related trends[26]. The combination of several shells with overlapping life-spans using cross-dating techniques allows one to obtain long-term and precisely dated records[23].

We explored changes in resilience through time in each record by assessing changes in lag-1 autocorrelation (AR(1)) and variance. Both metrics were computed by sliding a 70-year window over the residuals of the detrended time series, with the results plotted at the end of the sliding window (see Methods). Our analysis reveals the existence of two episodes in which variance and autocorrelation (AR(1)) increase together before 1380 CE (Fig. 2d, e). The first episode of declining resilience occurs between 1180 and 1260 (drawing on data from 1110 to 1260 due to the sliding window), spanning the last century and a half of the MCA. The second episode of declining resilience occurs between 1330 and 1380 (drawing on data from 1260 to 1380). Both episodes are separated by an interval that shows increase of resilience from 1260 to 1330 (drawing on data from 1190 to 1330). Although we can observe other periods after the second episode of declining resilience, in which either autocorrelation or variance increase, the trends are not clear, and there is no agreement between the two indicators or between proxies (Fig. 2d, e).

To measure the strength of the trends in the indicators for each period, we use the Kendall's $\tau$ correlation coefficient, in which positive values indicate an increasing trend, interpreted as declining resilience[29]. Significance is assessed by comparing the measured trend with the trends expected from a null model of surrogate time-series[29,30] (see Methods). Over the first interval, variance increases significantly in all records, whereas AR(1) exhibits a significant increasing trend in the shell-growth and $\delta^{18}O_{shell}$ series (Fig. 2 and Supplementary Fig. 1). In $\delta^{13}C_{shell}$, AR(1) experiences an abrupt jump around 1180 and then slightly decreases afterwards. Since AR(1) is a sensitive metric to abrupt changes in the input data, the negative spike observed around 1180 (Fig. 2b) probably influenced the trend in this indicator. Between both episodes, AR(1) and variance trends are negative and significant for some of the records (Supplementary Fig. 1), indicating increased stability. During the second episode, AR(1) in the $\delta^{18}O_{shell}$ series and variance in the $\delta^{13}C_{shell}$ record exhibit significant increasing trends (Fig. 2 and Supplementary Fig. 1), but none of the other metrics.

To define whether the observed trends are robust to the choice of sliding window size and filtering bandwidth, we measured the trends over a range of combinations of both parameters (see Methods). Declining resilience is more conspicuous during the first episode, with all records exhibiting positive Kendall $\tau$-values across most combinations of window lengths and detrending bandwidths (Fig. 3a). Variance exhibits positive and significant trends over most of the parameter space (median $\tau \geq 0.7$). AR(1) trends are more robust in the growth

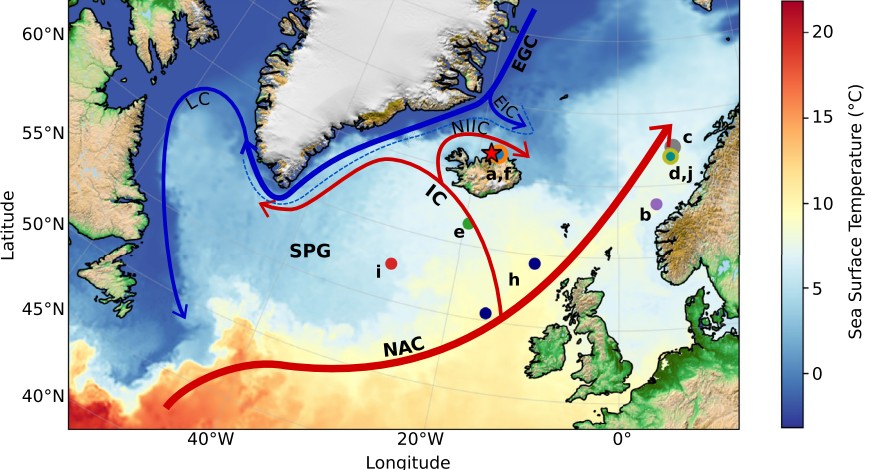

**Fig. 1 | Bivalve derived proxy records location and regional ocean circulation.** Red arrows represent the relatively warm and salty waters carried northwards by the North Atlantic Current (NAC), westward as the Irminger Current (IC), and clockwise around Iceland as the North Icelandic Irminger Current (NIIC). Blue arrows correspond to relatively cool and fresh Arctic waters transported southward by the East Greenland Current (EGC) that diverts north of Iceland as the East Iceland Current (EIC) and southwest of Greenland as the Labrador Current (LC). The subpolar gyre (SPG) flows anticlockwise in the subpolar North Atlantic. The dashed blue line represents the mean position of the North Atlantic Polar Front (PF). The red star indicates the location of the bivalve shell sampling site (80 m water depth). The coloured circles represent the locations of the paleoenvironmental reconstructions represented on Fig. 4: **a**, **f** correspond to reconstructions from North Iceland, **b**–**d**, **j** to records from the Norwegian margin, **e**, **i** to the subpolar region and **h** to the Northeast Atlantic. The colour map represents April sea surface temperatures during 2016, obtained from the ESA Climate Change Initiative data[64]. The land image was obtained from the NOAA National Oceanographic Data Center[65].

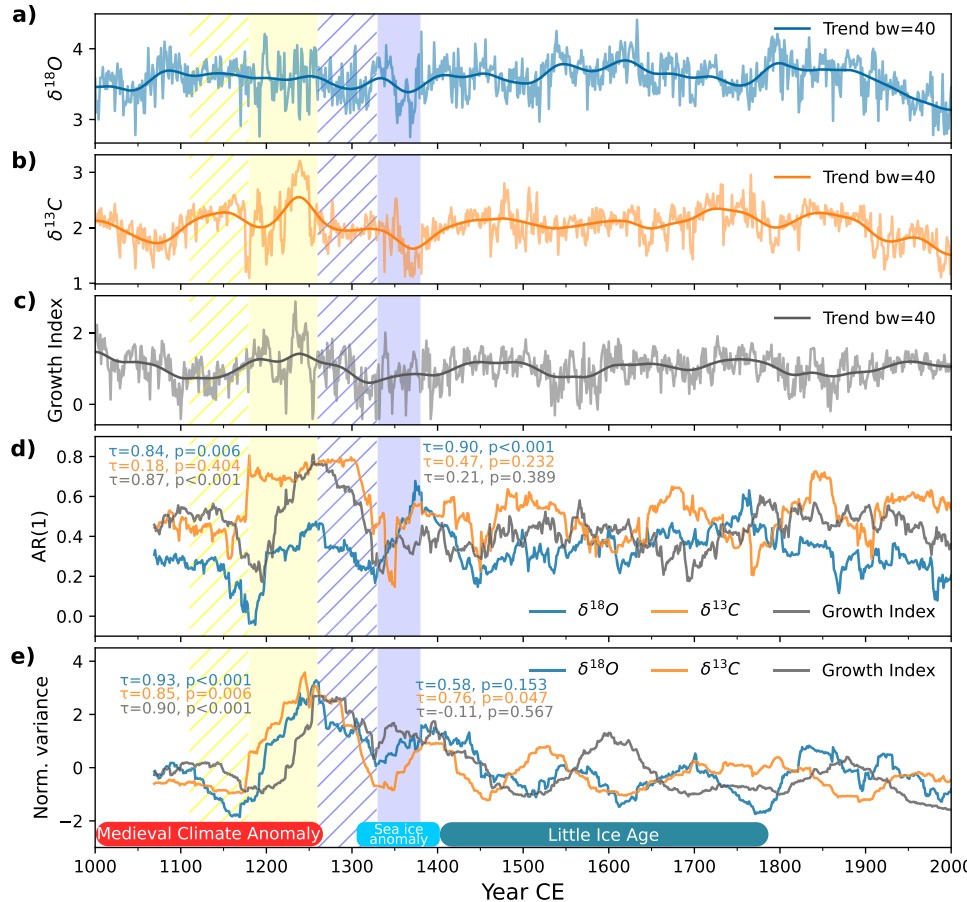

**Fig. 2 | Resilience changes measured along the bivalve shell-derived records.** **a**–**c** Annual values for the **a** $\delta^{18}O$[24], **b** $\delta^{13}C$[25], and **c** Growth Index[23] records. The thick lines represent the long-term trend estimated using a Gaussian smoothing kernel with a bandwidth of 40 years (see Methods). **d** AR(1) and **e** variance trends obtained using a window length of 70 years and a bandwidth of 40 years (see Methods). Declining resilience is detected over the yellow and blue shaded intervals corresponding to the first and second episodes respectively. The hatched region comprises the initial window interval for each episode.

index (median $\tau = 0.79$) and $\delta^{18}O_{shell}$ (median $\tau = 0.58$) records, whereas in $\delta^{13}C_{shell}$, AR(1) trends are slightly weaker (median $\tau = 0.4$) but still significant for a subset of combinations. The second episode is more robust in $\delta^{18}O_{shell}$ (median AR(1)-$\tau = 0.76$, var-$\tau = 0.56$), whereas it is less pronounced in $\delta^{13}C_{shell}$ (median AR(1)-$\tau = 0.26$, var-$\tau = 0.6$), and not evident in the shell-growth series (Fig. 3b). Positive trends in the indicators persist when the records are detrended using short bandwidths (Fig. 3 and Supplementary Fig. 2) indicating that the fluctuations exhibit intrinsic critical slowing down rather than autocorrelation arising from low frequency variability observed across each interval. This analysis confirms that the marine environment lost stability in two different episodes, the first having the strongest resilience loss signal occurred at the end of the MCA, the second during the transition into the LIA.

**Regional extent and oceanography of destabilisation**
The synchronicity across the records suggests that a common component of the system recorded in each proxy lost stability. Although shell-growth and $\delta^{13}C_{shell}$ integrate information on ecosystem processes, they are also influenced directly and indirectly by physical changes[23,25,26]. Furthermore, since $\delta^{18}O_{shell}$ contains information associated exclusively with physical processes[24,26], we can infer that the physical environment lost stability during both episodes. During the first interval, decreased resilience in shell-growth and $\delta^{13}C_{shell}$ indicates that the ecosystem tracked the environmental changes or even amplified them as suggested by the increased occurrence of extreme values over 1180–1260 (Fig. 2b, c). During the second episode, no

changes in resilience are observed in the shell-growth series. This latter interval coincides with the beginning of a period of increased export of Arctic sea ice into subpolar latitudes[31], including the North Icelandic Shelf[32,33] (Fig. 4a). The presence of sea ice likely disrupted primary production and food supply to the sea bed[34], as the observed decline in shell growth around 1300 suggests (Fig. 2c). Thus, whereas the isotope records were still sensitive to changes in the physical system, sea-ice driven ecosystem variability may have masked the physical environment's signal within shell growth records.

Comparisons with regional paleoenvironmental reconstructions can help to constrain the potential mechanisms driving the observed destabilisation and its geographical extent. $\delta^{13}C_{shell}$ and $\delta^{18}O_{shell}$ are sensitive to interannual sea surface temperature (SST) and surface salinity variability over the subpolar North Atlantic[24,25]. We therefore explore the results with respect to a range of proxy records from across the subpolar North Atlantic to determine the regional extent of the destabilisation.

Increasing autocorrelation and variance are usually found prior to abrupt transitions[20], including those that exhibit hysteresis once a tipping point is crossed[19,21]. Coincident with the increases in autocorrelation and variance we identify on the North Iceland Shelf, environmental reconstructions from across the subpolar North Atlantic experience rapid changes. SST reconstructions along the path of warm northward-flowing currents off Norway[35–38] and around Iceland[39–41] display a transitory cooling, followed by an episode of warmer conditions during the 14th century before a second and long-lasting cooling (Fig. 4b–f and Supplementary Fig. 3a). A similar pattern

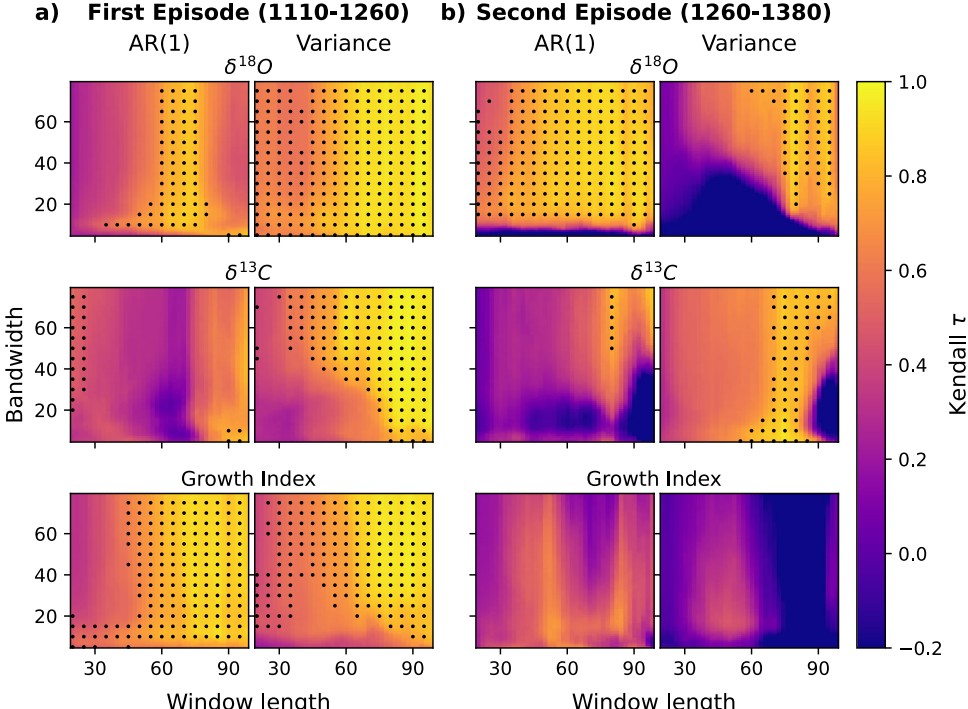

**Fig. 3 | Robustness analysis of window length and detrending bandwidth choice.** AR(1) and variance robustness during the **a** first and **b** second episodes for δ18Oshell (upper panels), δ13Cshell (middle panels) and Growth Index (lower panels). The colour maps represent the trend measured using Kendall τ coefficients, values close to 1 indicate loss of resilience. The combination of parameters that yield a significant trend ($p < =0.05$) are indicated by the black circles.

is seen in reconstructed Atlantic Multidecadal Variability[42] (Fig. 4g), implying impacts across the North Atlantic and linking the SST variability to changes in the strength of the North Atlantic circulation[43].

Modulation of northward heat transport by ocean circulation change[3,4,7] and abrupt SPG weakening have been proposed as drivers of the transition to the LIA cooling[8,18]. Reinforcing this hypothesis, the timing of destabilisation episodes identified in this study correspond with change in the influence of Labrador Sea Water on the northeastern Atlantic[44] (Fig. 4h), the strength of the Norwegian Atlantic Current[45] (Fig. 4j), and near-bottom flow speed of Iceland-Scotland Overturning Water in the subpolar North Atlantic[18] (Fig. 4i and Supplementary Fig. 3b), all linked to the strength of the SPG[46,47]. Despite dating uncertainties (Supplementary Table 1), these records exhibit a consistent pattern that suggests that the North Atlantic SPG weakened twice. Such a weakening of the gyre has been observed in modelling studies in response to a sea-ice driven dampening of subpolar North Atlantic convection[4,7,8]. The evidence presented here for loss of resilience in the subpolar North Atlantic before 1260, together with evidence for the weakening[18,44] of the potentially bistable[10,11,14] subpolar gyre, indicate that the onset of the LIA may have occurred in response to the subpolar gyre system passing a tipping point.

### Drivers of destabilisation

Our current understanding of the transition involves the activation of coupled sea-ice ocean feedbacks that followed an increased export of Arctic sea-ice into the subpolar North Atlantic[3–5,7]. This hypothesis implies synchronicity between intervals with high concentrations of sea ice and the destabilisation episodes that reflect the action of positive feedbacks. Several reconstructions record a period of significant export of Arctic sea ice into the North Atlantic during the 14th century[31]. However, the first destabilisation episode occurred before this, indicating that the North Atlantic was already losing stability. During the first interval, proxy records from southeast Greenland[48,49] and the Labrador Sea[50] indicate that surface conditions were anomalously fresh towards the end of the MCA. Fresh ice-laden waters in this

area can be sourced either from the Arctic, transported by the East Greenland Current, or from melting glaciers[49]. Lithological fingerprinting suggests that the ice rafted debris found within this period was sourced in south Greenland, potentially generated by glacier calving during the MCA warm conditions[49–51]. The strong signal of loss of stability we observe was therefore potentially associated with freshwater input into the subpolar North Atlantic, derived from glacier melting. The second interval is synchronous with the increased export of Arctic sea-ice into the North Atlantic during the 14th century[31,32] (Fig. 4a), and is consistent with the proposition of a sea-ice mediated amplification of an initial transient cooling.

There is an ongoing debate regarding the role of external forcing in the MCA-LIA transition. Some numerical studies indicate that the cooling effect of frequent strong volcanic eruptions, starting with the Samalas eruption in 1257, led to the expansion of Arctic sea ice that was exported later into subpolar latitudes[3,4,6]. An alternative view suggests that during periods of solar minima, the westerly winds that drive the subpolar gyre are diverted or blocked by the development of a persistent high-pressure system in the Northeast Atlantic[41,52]. This condition has also been suggested to strengthen the northeasterly winds, flushing sea ice from the Arctic into the subpolar North Atlantic[53], reinforcing the weakening of the SPG. Abrupt climate cooling in response to increased export of Arctic sea-ice has also been observed in unforced climate simulations[54–56]. Although the first destabilisation episode is coeval with a period of frequent volcanic activity[57](Fig. 4l), it occurred before land records showed signs of the initial cooling that is thought to have triggered the Arctic sea-ice expansion[2,3]. This evidence suggests that it is unlikely that the initial destabilisation occurred in response to short-term external forcing, and instead was a non-linear response to anomalous Medieval warmth. As for the second episode, it is possible that the cooling effect caused by the strong volcanic eruptions[57] and the Wolf solar minimum during the second half of the 13th century (Fig. 4k, l) had contributed to the expansion of Arctic sea-ice[31], which was later exported into subpolar latitudes and its melting potentially enhanced by the warm episode during the 14th century[43].

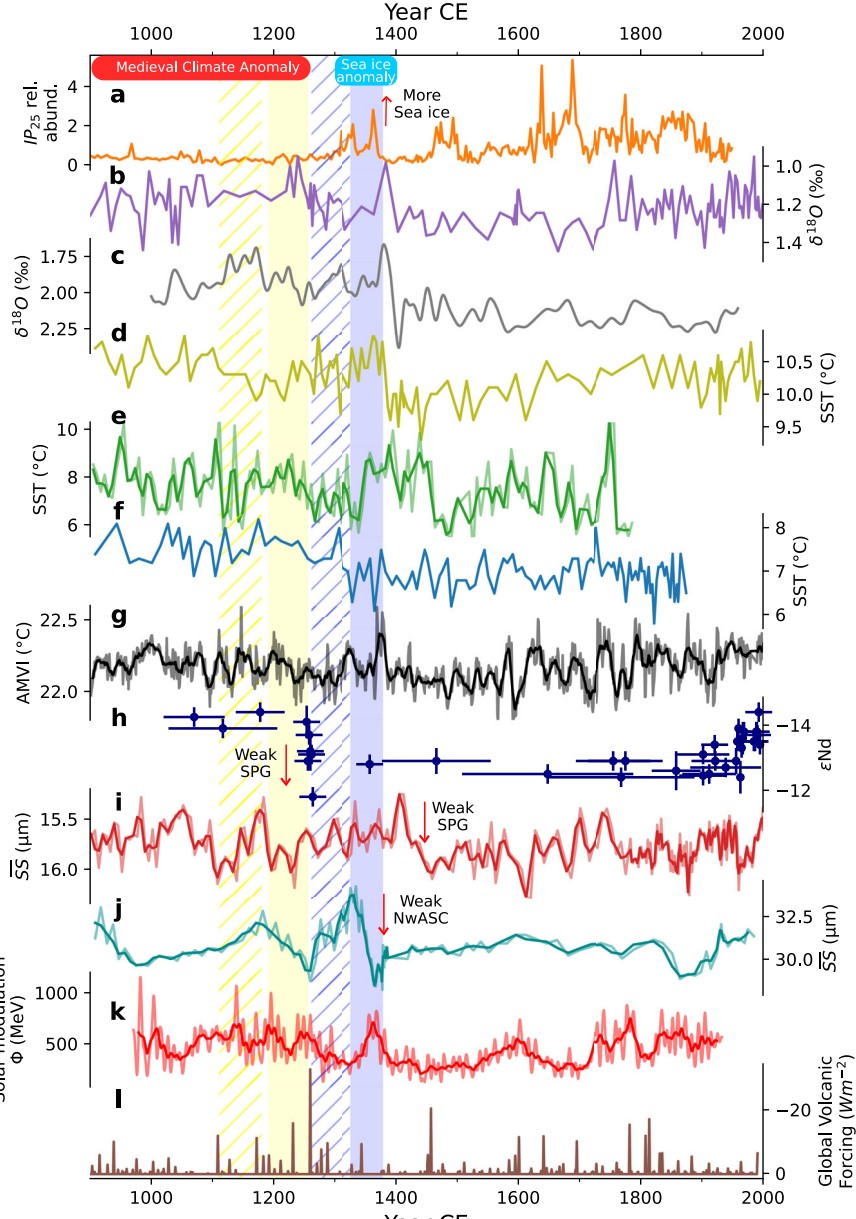

**Fig. 4 | Regional environmental reconstructions spanning the last millennium.**
**a** April sea ice index from the North Icelandic Shelf based on the relative abundance of IP25, a biomarker synthesised by sea-ice diatoms[32]. **b** Oxygen isotope record from the planktonic foraminifera *Neogloboquadrina pachyderma* reflecting summer near-surface temperatures on the Norwegian margin[35]. **c** Oxygen isotope record as a proxy for SSTs on the Norwegian margin[36]. **d** August SST on the Eastern Norwegian Sea inferred from diatom assemblages[38]. **e** Annual SST in South Iceland based on paired Mg/Ca–$\delta^{18}$O measurements from the planktonic foraminifera *Globorotalia inflata*[41]. **f** Summer SST in North Iceland based on diatom assemblages[39]. **g** Atlantic Multidecadal Variability Index (AMVI) reconstructed from Titanium content from a laminated sedimentary record from the Canadian Arctic[42]. **h** Neodymium isotopic composition ($\varepsilon$Nd) in deep-sea corals from the north-eastern Atlantic[44], dating uncertainties are indicated by the horizontal error bars. Higher $\varepsilon$Nd isotopic values indicate a decreased influence of Labrador Sea Water. **i** Iceland-Scotland Overturning Water (ISOW) sortable silt average size ($\overline{SS}$)[18], faster flows indicate a weakened influence of Labrador Sea water. **j** Sortable silt average size ($\overline{SS}$) as a proxy for variations in the strength of the Norwegian Atlantic Slope Current (NwASC)[45]. **k** Reconstructed solar modulation parameter ($\Phi$) as a proxy for solar activity[62]. **l** Global volcanic forcing reconstructed from sulfate composite ice core records[57]. The bold lines in some of the records were obtained by computing 3 to 20 year moving averages to improve visibility. The yellow and blue shaded intervals correspond to the first and second destabilisation events respectively. The hatched region comprises the initial window interval for each episode. The interval comprising the sea ice anomaly event is represented by the blue square.

Our analysis indicates that the subpolar North Atlantic system experienced a two-phase destabilisation during the MCA-LIA transition that likely reflect the action of sea-ice to ocean-atmosphere feedbacks. The most noticeable episode occurred before the initial cooling that is hypothesised to have triggered the LIA, and potentially culminated in an abrupt weakening of the subpolar gyre, increasing the system vulnerability to further perturbations. The vulnerability of the North Atlantic system to externally forced perturbations is a critical issue today, with recent analysis suggesting that this system has destabilised during the last century and might be approaching a tipping point[15]. If rapid loss of Arctic sea-ice[58], accelerating melting of the Greenland ice sheet[59], and associated export of freshwater into key convective regions in the North Atlantic[60] continues, a subpolar gyre tipping point[11,12,14] might again lead to rapid and long-lasting regional climate change.

## Methods

### Datasets

We obtained the bivalve-derived proxies from the NOAA National Centers for Environmental Information (NCEI) Paleoclimatology repository. We used the raw data for the calculation of the resilience metrics.

### Calculation of variance and autocorrelation trends

To assess changes in the pattern of fluctuations that may indicate loss of stability, we computed the resilience metrics on the residuals after detrending each record. The trend was computed as the moving average weighted by a Gaussian kernel function with a prescribed bandwidth[61]. Variance and lag-1 autocorrelation were calculated within a rolling window that moves one point at a time over the residuals series. Lag-1 autocorrelation is obtained by successively fitting a first-order autoregressive model (AR(1)) using an ordinary least-squares method[61]. The AR(1) model has the form: $y_{t+1} = \alpha y_t + \sigma \varepsilon$, where $t$ represents time, $y$ the portion of the time series within the window, $\alpha$ is the AR(1) coefficient, and $\varepsilon$ represents white noise with variance $\sigma^2$. To assess the trend in the indicators, we use the non-parametric Kendall's $\tau$ rank correlation coefficient to cross-correlate time and the indicator series[29]. The resulting Kendall $\tau$ coefficient ranges between −1 and 1: where values close to 1 indicate an increasing trend, and −1 a decreasing trend. Sensitivity to different combinations of window lengths and filtering bandwidths is represented as colour maps of Kendall $\tau$ values[29].

### Significance tests

To test the significance of the trends in the indicators, we compared the trend measured in each record with the expected trends from a null model[29]. The model consists of 3000 surrogate series, each generated using a bootstrapping method by sampling with replacement from the residuals of the original time-series to ensure they have the same spectrum as the original records[30]. We estimated Kendall $\tau$ trends for each surrogate series after computing AR(1) and variance using the same window length and bandwidth as in the original series. Lastly, we define the $p$-value as the proportion of series that exhibit a $\tau$-value greater than or equal to that observed in the original series[30]. This process was repeated for a subset of combinations of window length and bandwidth uniformly distributed over the parameter space.

### Selection of the environmental reconstructions

The temperature proxy records included in this study were selected according to these criteria: (a) records located along the path of the northward flowing currents, (b) selection of isotope-based reconstructions when possible to reduce some of the intrinsic proxy biases, (c) high-temporal average resolution over the MCA-LIA transition period (<12 years), and (d) dating uncertainties less than the duration of each destabilisation episode. The proxy records for circulation changes are less numerous, thus most of those that lie on the path of the northward flowing currents were included. The external forcing records selected were the latest available reconstructions.

The oxygen isotope record by Sejrup et al.[35] was retrieved on 7 March 2022 from https://www.ncei.noaa.gov/access/paleo-search/study/22592. The oxygen isotope record by Andersson et al.[36] was accessed on 10 March 2022 at https://www.ncei.noaa.gov/access/paleo-search/study/14193. August SST from the Norwegian margin[38] was obtained on 26 August 2021 from https://www.ncei.noaa.gov/access/paleo-search/study/17475. Annual SST from South Iceland[41] was accessed on 7 March 2022 at https://www.ncei.noaa.gov/access/paleo-search/study/19804. Summer SST from North Iceland[39] was retrieved on 17 June 2021 from https://www.ncei.noaa.gov/access/paleo-search/study/17635. The AMVI reconstruction[42] was accessed on 7 March 2022 from https://www.ncei.noaa.gov/access/paleo-search/study/31353. The sortable silt record from the subpolar North Atlantic[18] was retrieved on 26 August 2021 from https://www.ncei.noaa.gov/access/

paleo-search/study/22790. Neodymium measurements from the Rockall Trough were obtained from the original study[44]. The solar activity[62] and volcanic radiative forcing[57] records were obtained from the original studies on 10 March 2022.

## Data availability

The $\delta^{18}O$ record[24] used in this study was retrieved in July 2020 from https://www.ncei.noaa.gov/access/paleo-search/study/20448. The $\delta^{13}C$ series[25] was accessed in January 2021 at https://www.ncei.noaa.gov/access/paleo-search/study/22950. The Growth Index record[23] is available at https://www.ncei.noaa.gov/access/paleo-search/study/14609.

## Code availability

All python code used for the analyses is available on the repositories: https://zenodo.org/record/6856039[63] and https://github.com/BeatrizArellano/regimeshifts.

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

## Acknowledgements

B.A.N. and P.H. were funded by the European Union's Horizon 2020 research and innovation programme under grant agreement No 820989 (project COMFORT, Our common future ocean in the Earth system – quantifying coupled cycles of carbon, oxygen, and nutrients for determining and achieving safe operating spaces with respect to tipping points). The work reflects only the authors' view; the European Commission and their executive agency are not responsible for any use that may be made of the information the work contains. P.H., J.S., and D.J.R. were funded by the NERC grant NE/N001435/1. C.A.B. and T.M.L. were supported by the Leverhulme Trust grant RPG-2018-046. C.A.B. also acknowledges support from the 'Tipping Points in the Earth System' (TiPES) programme. This project is TiPES contribution #133, and has received funding from the European Union's Horizon 2020 research and innovation programme under grant agreement No 820970. P.B., J.S. and D.J.R. were funded by the European Research Council (ERC) under the European Union's Horizon 2020 research and innovation programme SEACHANGE (grant agreement No 856488). The authors would like to thank Simon T. Belt for providing the sea-ice reconstruction from North Iceland.

## Author contributions

P.R.H. and B.A.N. conceived and led the research. C.A.B. and T.M.L. helped to shape the research. P.G.B., D.J.R. and J.S. contributed with data and ideas for the analyses. B.A.N. performed the analysis. B.A.N., P.R.H., T.M.L, C.A.B., J.S., P.G.B., and D.J.R. contributed to the writing of the manuscript.

## Competing interests

The authors declare no competing interests.

## Additional information

**Supplementary information** The online version contains

supplementary material available at https://doi.org/10.1038/s41467-022-32653-x.

