## [Peer Review File · Nature Communications]

Destabilisation of the Subpolar North Atlantic prior to the Little Ice AgeReviewers' Comments:

Reviewer #1:

Remarks to the Author:

Review of "Destabilisation of the North Atlantic prior to the Little Ice Age"

by Beatriz Arellano-Nava, Paul R. Halloran, Chris A. Boulton, James Scourse, Paul G. Butler, David J. Reynolds, and Timothy M. Lenton

General comments:

The paper presents an empirical statistical analysis of three existing ultra-high resolution sclerochronological records from the north Iceland shelf, a sensitive key region in the northern North Atlantic, then compared with several marine reconstructions from the north Iceland shelf and other regions. It addresses the important and timely topic of understanding abrupt changes and tipping points in the coupled ocean-climate system, here focused on major and abrupt transitions associated with the Little Ice Age. This is a significant advancement. It is a creative and novel analysis that leverages these unique records well beyond the impressive previous analyses by these authors and others. It appears technically sound, well-justified and well-explained – and the results and the conclusion are supported by the evidence presented. The paper, including the description of resilience and its statistical indicators (from lines 73-130) represents a thorough and convincing advance upon the authors' 2021 EGU conference presentation (EGU21-16401). I think that this is an outstanding paper and is meritorious enough for publication in a high-visibility journal such as Nature Communications. I recommend it for publication with only minor revisions – only the first or second time as a reviewer that I have recommended only minor revisions. (Hence the brevity of my comments, which are limited despite having read this manuscript multiple times.)

Specific comments:

- The issue of what may have caused the destabilization in the late 12th and early 13th centuries seems to be left largely untouched. Along with this, the whole issue of external forcing (volcanic and/or solar) as a driver is also side stepped. I would like to see some discussion of this in the paper.

There are some minor concerns with the description / presentation in the manuscript.

- Pg. 7, lines 168-174: "... sea ice was sourced .. due to glacier calving." The term "sea ice" is not the same as "ice in the sea" and should be reserved for frozen seawater, whether landfast or drifting and whether seasonal or multiyear ice from the Arctic Ocean. The authors should at least re-phrase these sentences – and even re-consider the interpretation. Personally, I have problems with the interpretation of Alonso-Garcia et al. 2017 (reference 37).
- Pg. 16, Fig. 3 caption. I suggest adding a sentence about "... d18O (upper panels), d13C (middle panels) and growth index (lower panels).
- Pg. 17, Fig. 4. The two SST plots (Fig. 4b and 4c) are inconsistent in their sign, with Fig. 4d shown as inverted values – and not stated in the caption. These two should be the same, whether both or neither inverted. It is the authors' choice, but these should be consistent. I suggest adding some text and arrows in the figure itself, similar to reference 31 (their Figs. 2-4), which also compared sea ice and SST as interpreted from various proxies.
- Supplementary information: Pg. 1, Fig. S1 caption. Same as the comment about pg. 16, Fig.3: I suggest adding a sentence about "... d18O (upper panels), d13C (middle panels) and growth index (lower panels).

Reviewer #2:

Remarks to the Author:

This is an interesting study which attempts to document periods of destabilization in the North Atlantic prior to the Little Ice Age. The analysis is based on three variables ($\delta^{18}O$, $\delta^{13}C$, and width growth) on bivalve shell records from the Northern Icelandic basin. This location is in the confluence of many components of the broader north Atlantic current: the Irminger current, the East Greenland Current, the East Iceland Current (as well as the subpolar gyre). The authors then measured trends in both variance and temporal autocorrelation to assign loss of resilience of the climate system in that area. Overall, I found the paper easy to read and may provide new insights into destabilization of the North Atlantic prior to the Little Ice Age. However, before I can recommend this paper to be published in Nature Communications, there are some aspects that need to be further discussed.

First of all, I am not totally convinced about the selection of the proxies in Fig. 4, there needs to be some criteria as to why the authors chose these specific sites (there are many others available). A better proxy than Nd isotopic values for the influence of Labrador Sea is probably the sortable silt located south of Iceland from Moffa-Sanchez and Hall (2017). This proxy's temporal resolution is extremely high (~ 6 years), I am wondering why it is not in the figure. Also, as the title currently reads, it is expected the whole North Atlantic was destabilized, but given the location of the study area (North Iceland shelf) and the various currents affecting it, the destabilization may be more related to the Nordic Seas?

Secondly, the two peaks of lower $\delta^{18}O$ in the late 1300s seem to be in phase with a recent reconstructed Atlantic multidecadal oscillation (Lapointe and Bradley, 2021, Fig.3a). The AMV record from Mann et al. (2009) also depicts two warmth peaks at the end of the 14th century, which supports the hypothesis of a possible stronger North Atlantic current during that time. The record of SST in the Vøring Plateau off Norway (this study, Fig. 4e) is also shown in Fig. 2A from Lapointe and Bradley (2021), together with many other highly resolved proxies. This poleward heat transfer seems to be the precursor of the "Great Sea ice anomaly" as defined by Miles et al. (2020). Hence, this does suggest a flux of warm water into the Nordic Seas during that period, followed by a weakening of northward heat transport. In this regard, lines 184-185 "this suggests that northward heat transport weakened rapidly after each episode" make a lot of sense, especially for the second destabilization period.

Similarly, the first period also appears to have been one of increased sea-ice in the subpolar gyre (Labrador Sea) due to the "warmer" MCA period triggering calving and sea ice export (lines 171-174). One question raised: Are these two episodes of "declining resilience" caused by warming SST? It is not explicitly put forward in the current manuscript, but it seems to be plausible.

The "weakening of negative feedbacks" may be linked with the accumulation of arctic sea ice in the arctic high latitudes during the late 1300s (Miles et al. 2020). This, in turn, may be consistent with persistent atmospheric blocking over Greenland in times of warmer SST in the subpolar gyre as shown by many proxies in the Labrador Sea, as was the case in the 1960s (Ionita et al. 2016, Lapointe and Bradley 2021).

As for the interpretation of the second "destabilization" period: the two peaks in very low values of $\delta^{18}O$ at the end of the 1300s is in phase with the significant AR(1) ($\tau = 0.90$, $p < 0.001$). Wouldn't that be expected that autocorrelation in the $\delta^{18}O$ data be large during this period of increasingly lower values (Reynolds et al. 2016)? I would like the authors comment on that.

Minor comments:

Figure 1: indicate all the letters (sites name) in the caption (a to e).

Line 64: three annually resolved proxy records. These proxies are d18O, d13C and band width from the same location. Am I right?

Line 99: Do you imply that 1400 was the onset of the Little Ice Age? If so, it would be probably good to define it in the introduction.

Lines 181-184:

The argument of a rapid drop in SST following the first and especially the second destabilization events is not very convincing for the summer SST reconstruction based on alkenone palaeothermometry (Fig. 4b), i.e., the SST drop for this proxy occurs very much in phase with the second destabilization event. Also, the drop of SST in the Vøring Plateau is not too obvious after the first episode.

Lines 196-197: the initial cooling may have been caused by warming conditions during the MCA triggering calving and sea ice export. It would be needed to also discuss the 1257 Samalas eruption and the possible positive (sea ice) feedbacks associated with it.

References (not mentioned in the main text)

Ionita, Monica, et al. "Linkages between atmospheric blocking, sea ice export through Fram Strait and the Atlantic Meridional Overturning Circulation." *Scientific reports* 6.1 (2016): 1-10.

Moffa-Sánchez, Paola, and Ian R. Hall. "North Atlantic variability and its links to European climate over the last 3000 years." *Nature communications* 8.1 (2017): 1-9.

Lapointe, Francois, and Raymond S. Bradley. "Little Ice Age abruptly triggered by intrusion of Atlantic waters into the Nordic Seas." *Science advances* 7.51 (2021): eabi8230.

Response to Reviewers

We thank the referees for their valuable and insightful comments and suggestions. In line with the reviewers' comments the revised manuscript presents the same data and analysis, and reaches the same overarching conclusions as the original submission, but expands considerably the discussion and context in which these results are placed. We reproduce the comments and answer to each point below, referring to substantive changes made to the revised manuscript.

Reviewer #1 (Remarks to the Author):

Review of “Destabilisation of the North Atlantic prior to the Little Ice Age”

by Beatriz Arellano-Nava, Paul R. Halloran, Chris A. Boulton, James Scourse, Paul G. Butler, David J. Reynolds, and Timothy M. Lenton

General comments:

The paper presents an empirical statistical analysis of three existing ultra-high resolution sclerochronological records from the north Iceland shelf, a sensitive key region in the northern North Atlantic, then compared with several marine reconstructions from the north Iceland shelf and other regions. It addresses the important and timely topic of understanding abrupt changes and tipping points in the coupled ocean-climate system, here focused on major and abrupt transitions associated with the Little Ice Age. This is a significant advancement. It is a creative and novel analysis that leverages these unique records well beyond the impressive previous analyses by these authors and others. It appears technically sound, well-justified and well-explained – and the results and the conclusion are supported by the evidence presented. The paper, including the description of resilience and its statistical indicators (from lines 73-130) represents a thorough and convincing advance upon the authors' 2021 EGU conference presentation (EGU21-16401). I think that this is an outstanding paper and is meritorious enough for publication in a high-visibility journal such as Nature Communications. I recommend it for publication with only minor revisions – only the first or second time as a reviewer that I have recommended only minor revisions. (Hence the brevity of my comments, which are limited despite having read this manuscript multiple times.)

We thank the reviewer for the kind and very positive comments around timeliness, creativity, novelty and quality of this work.

Specific comments:

1. The issue of what may have caused the destabilization in the late 12th and early 13th centuries seems to be left largely untouched. Along with this, the whole issue of external forcing (volcanic and/or solar) as a driver is also side stepped. I would like to see some discussion of this in the paper.

Thank you for pointing out this gap in our discussion. We have elaborated on the potential drivers behind the first destabilisation episode and the role of

external forcing in the new subsection “Drivers of destabilisation” in the Discussion (page 8, lines 187-225). Here we reproduce those paragraphs:

“Our current understanding of the transition involves the activation of coupled sea-ice ocean feedbacks that followed an increased export of Arctic sea-ice into the subpolar North Atlantic^{3-5,7}. This hypothesis implies synchronicity between intervals with high concentrations of sea ice and the destabilisation episodes that reflect the action of positive feedbacks. Several reconstructions record a period of significant export of Arctic sea ice into the North Atlantic during the 14th century³¹. However, the first destabilisation episode occurred before this, indicating that the North Atlantic was already losing stability. During the first interval, proxy records from southeast Greenland^{48,49} and the Labrador Sea⁵⁰ indicate that surface conditions were anomalously fresh towards the end of the MCA. Fresh ice-laden waters in this area can be sourced either from the Arctic, transported by the East Greenland Current, or from melting glaciers⁴⁹. Lithological fingerprinting suggests that the ice rafted debris found within this period was sourced in south Greenland, potentially generated by glacier calving during the MCA warm conditions⁴⁹⁻⁵¹. The strong signal of loss of stability we observe was therefore potentially associated with freshwater input into the subpolar North Atlantic, derived from glacier melting. The second interval is synchronous with the increased export of Arctic sea-ice into the North Atlantic during the 14th century^{31,32} (Fig. 4a), and is consistent with the proposition of a sea-ice mediated amplification of an initial transient cooling.

There is an ongoing debate regarding the role of external forcing in the MCA-LIA transition. Some numerical studies indicate that the cooling effect of frequent strong volcanic eruptions, starting with the Samalas eruption in 1257, led to the expansion of Arctic sea ice that was exported later into subpolar latitudes^{3,4,6}. An alternative view suggests that during periods of solar minima, the westerly winds that drive the subpolar gyre are diverted or blocked by the development of a persistent high-pressure system in the Northeast Atlantic^{41,52}. This condition has also been suggested to strengthen the northeasterly winds, flushing sea ice from the Arctic into the subpolar North Atlantic⁵³, reinforcing the weakening of the SPG. Abrupt climate cooling in response to increased export of Arctic sea-ice has also been observed in unforced climate simulations⁵⁴⁻⁵⁶. Although the first destabilisation episode is coeval with a period of frequent volcanic activity⁵⁷(Fig. 4I), it occurred before land records showed signs of the initial cooling that is thought to have triggered the Arctic sea-ice expansion^{2,3}. This evidence suggests that it is unlikely that the initial destabilisation occurred in response to short-term external forcing, and instead was a non-linear response to a gradual relaxation of anomalous Medieval

warmth. As for the second episode, it is possible that the cooling effect caused by the strong volcanic eruptions⁵⁷ and the Wolf solar minimum during the second half of the 13th century (Fig. 4k,l) had contributed to the expansion of Arctic sea-ice³¹, which was later exported into subpolar latitudes and its melting potentially enhanced by the warm episode during the 14th century⁴³.”

There are some minor concerns with the description / presentation in the manuscript:

2. Pg. 7, lines 168-174: “... sea ice was sourced .. due to glacier calving.” The term “sea ice” is not the same as “ice in the sea” and should be reserved for frozen seawater, whether landfast or drifting and whether seasonal or multiyear ice from the Arctic Ocean. The authors should at least re-phrase these sentences – and even re-consider the interpretation. Personally, I have problems with the interpretation of Alonso-Garcia et al. 2017 (reference 37).

Thank you for highlighting this wording error, we have rephrased that sentence (page 8, lines 197-199) and broadened the interpretation. We discuss the possibility that the ice rafting debris to the South of Greenland towards the end of the Medieval Climate Anomaly (MCA) was sourced from glacier calving. We have further reviewed the literature and a larger number of proxy records to assess the potential drivers of the first destabilization episode. To the best of our knowledge, towards the end of the MCA, anomalously fresh conditions were only reported around south Greenland and the Labrador Sea. Evidence for this, together with the suggestion that the ice was sourced in Southeast Greenland potentially due to glacier calving has been proposed by at least three different studies (Alonso-Garcia et al., 2017; Andresen et al., 2013; Andrews & Jennings, 2014; Miettinen et al., 2015). As there may be other potential explanations, we consider this to be a plausible mechanism behind the first destabilization episode as it is congruent with the weakening of the subpolar gyre that may have occurred after this event as indicated by some other environmental reconstructions. We present this hypothesis briefly in the revised manuscript (lines 188 to 205).

3. Pg. 16, Fig. 3 caption. I suggest adding a sentence about “... d18O (upper panels), d13C (middle panels) and growth index (lower panels).

Thank you for this useful suggestion. It has been implemented in Figure 3 caption (Page 21, lines 360-361).

4. Pg. 17, Fig. 4. The two SST plots (Fig. 4b and 4c) are inconsistent in their sign, with Fig. 4d shown as inverted values – and not stated in the caption. These two should be the same, whether both or nether inverted. It is the authors’ choice, but these

should be consistent. I suggest adding some text and arrows in the figure itself, similar to reference 31 (their Figs. 2-4), which also compared sea ice and SST as interpreted from various proxies.

Thank you for pointing out this inconsistency, we have now ensured that the reconstructions in Figure 4 are represented appropriately. Following a separate comment, we have reviewed a larger number of records and included some of them in the figure. Also, we have implemented your suggestion and included text and arrows to highlight the important changes that occurred throughout the transition.

5. Supplementary information: Pg. 1, Fig. S1 caption. Same as the comment about pg. 16, Fig.3: I suggest adding a sentence about "... d18O (upper panels), d13C (middle panels) and growth index (lower panels).

Thank you again for the suggestion, it has been implemented in Figure S1 caption (Supplementary Information, Page 1 lines 6-7).

Reviewer #2 (Remarks to the Author):

This is an interesting study which attempts to document periods of destabilization in the North Atlantic prior to the Little Ice Age. The analysis is based on three variables (d18O, 13C, and width growth) on bivalve shell records from the Northern Icelandic basin. This location is in the confluence of many components of the broader north Atlantic current: the Irminger current, the East Greenland Current, the East Iceland Current (as well as the subpolar gyre). The authors then measured trends in both variance and temporal autocorrelation to assign loss of resilience of the climate system in that area. Overall, I found the paper easy to read and may provide new insights into destabilization of the North Atlantic prior to the Little Ice Age. However, before I can recommend this paper to be published in Nature Communications, there are some aspects that need to be further discussed.

We thank the reviewer and are pleased that they found the manuscript easy to read and saw potential new insights into the destabilisation prior to the Little Ice Age. We have embraced the encouragement to expand the discussion of aspects of the manuscript.

1. First of all, I am not totally convinced about the selection of the proxies in Fig. 4, there needs to be some criteria as to why the authors chose these specific sites (there are many others available). A better proxy than Nd isotopic values for the influence of Labrador Sea is probably the sortable silt located south of Iceland from Moffa-Sanchez and Hall (2017). This proxy's temporal resolution is extremely high (~6 years), I am wondering why it is not in the figure.

Thank you for this suggestion. We have now undertaken a comprehensive review of environmental reconstructions from the region and established criteria for their selection. We have incorporated a subsection “Selection of the environmental reconstructions” in the Methods (pages 11-12, lines 275-299) and modified Figure 4 in the main text. In the new selection of reconstructions, we have included the sortable silt record by Moffa-Sánchez and Hall (2017). We have also carefully assessed the suitability of the Neodymium isotope record included in the main text. We consider it to be a particularly valuable reconstruction, despite its low resolution, because the fossil corals archive allows for U-Th dating with narrow uncertainties (± 22 years) during the initial destabilization period.

2. Also, as the title currently reads, it is expected the whole North Atlantic was destabilized, but given the location of the study area (North Iceland shelf) and the various currents affecting it, the destabilization may be more related to the Nordic Seas?

This is a good point, and we have modified the title to specify ‘subpolar North Atlantic’. We also use the extended compilation of proxy reconstructions undertaken to address the previous question to argue that destabilisation across the subpolar gyre, Labrador and Irminger Seas occurs (revised figures 1 and 4), recognizing also that it is a system with evidence for a tipping point (pages 7-8, lines 154-185).

3. Secondly, the two peaks of lower $d_{18}O$ in the late 1300s seem to be in phase with a recent reconstructed Atlantic multidecadal oscillation (Lapointe and Bradley. 2021, Fig.3a). The AMV record from Mann et al. (2009) also depicts two warmth peaks at the end of the 14th century, which supports the hypothesis of a possible stronger North Atlantic current during that time. The record of SST in the Vøring Plateau off Norway (this study, Fig. 4e) is also shown in Fig. 2A from Lapointe and Bradley (2021), together with many other highly resolved proxies. This poleward heat transfer seems to be the precursor of the “Great Sea ice anomaly” as defined by Miles et al. (2020). Hence, this does suggest a flux of warm water into the Nordic Seas during that period, followed by a weakening of northward heat transport. In this regard, lines 184-185 “this suggests that northward heat transport weakened rapidly after each episode” make a lot of sense, especially for the second destabilization period.

Thank you for this insightful observation and for highlighting the recent evidence for the strengthening of the North Atlantic Circulation during the 14th century. Reinforcing the point made by the reviewer, we have expanded the set of reconstructions discussed in the manuscript to demonstrate the wealth of evidence behind the existence of two periods of weakened North Atlantic circulation (pages 7-8, lines 154-185; revised figure 4). Both events are interrupted by the warm episode detected by Lapointe and Bradley (2021) as highlighted by the reviewer. Specifically:

- We see the first cooling event in the Atlantic Multidecadal Variability reconstruction (Fig. 4g). In this record, the cold interval spans part of the 13th century and is comparable in magnitude with the observed cooling after the year 1400 CE.
- An oxygen isotope record from the Norwegian margin (Fig. 4b), which is exceptionally well-dated (± 20 years; see Table S1) captures a rapid cooling around 1260 followed by the warm episode during the 14th century that ended abruptly around 1400 CE.
- Two further temperature reconstructions from the Norwegian margin (Figs. 4c and S3a) and South Iceland (Fig. 4e) display a cooling event around 1260 (within their dating uncertainties).
- Circulation change during these intervals is also indicated in the sortable silt record by Moffa-Sánchez and Hall (2017) (Fig. 4i), which has been interpreted as a proxy for the SPG strength. This record indicates a transitory SPG weakening within dating uncertainty (between ± 41 and ± 71 years) of the 1260 event.
- Sortable silt data from South Iceland (same location as record e on Fig. 1), interpreted as a proxy for the speed of the Iceland-Scotland Overflow Waters (ISOW) (Moffa-Sanchez et al., 2015), exhibits an abrupt weakening around 1250 (Fig. S3b). This weakening may be attributed to an abrupt decrease of the Atlantic inflow into the Nordic Seas. The ISOW weakening may also explain why the sortable silt record downstream Moffa-Sánchez and Hall (2017) (Fig. 4i) does not exhibit a more conspicuous weakening during the first episode, as this proxy is influenced by the relative strength of the ISOW and Labrador Sea Water formation.
- Finally, a proxy for the relative influence of Atlantic water in the eastern Labrador Sea based on the difference in the oxygen isotope signal of two planktonic foraminifera species (Fig. S4c) exhibits a transitory decrease in the influence of Atlantic water around 1260 with a recovery during the 14th century and an abrupt and longer-lasting weakening around 1400.

In summary, northward heat transport appears to have weakened rapidly after each hypothesized destabilisation episode, the first weakening being transitory, whereas the second one was longer-lasting. Here we reproduce the changes made to the manuscript (page 7, lines 161-185):

“Increasing autocorrelation and variance are usually found prior to abrupt transitions²⁰, including those that exhibit hysteresis once a tipping point is crossed^{19,21}. Coincident with the increases in autocorrelation and variance we identify on the North Iceland Shelf, environmental reconstructions from across the subpolar North Atlantic experience rapid changes. SST reconstructions along the path of warm northward-flowing currents off Norway^{35–38} and around Iceland^{39–41} display a transitory cooling, followed by an episode of warmer conditions during the 14th century before a second and long-lasting cooling (Figs. 4b-f and Sa). A similar pattern is seen in reconstructed Atlantic Multidecadal Variability⁴² (Fig. 4g), implying

impacts across the North Atlantic and linking the SST variability to changes in the strength of the North Atlantic circulation⁴³.

Modulation of northward heat transport by ocean circulation change^{3,4,7} and abrupt SPG weakening have been proposed as drivers of the transition to the LIA cooling^{8,18}. Reinforcing this hypothesis, the timing of destabilisation episodes identified in this study correspond with change in the influence of Labrador Sea Water on the north-eastern Atlantic⁴⁴ (Fig. 4h), the strength of the Norwegian Atlantic Current⁴⁵ (Fig. 4j), and near-bottom flow speed of Iceland-Scotland Overturning Water in the subpolar North Atlantic¹⁸ (Fig. 4i), all linked to the strength of the SPG^{46,47}. Despite dating uncertainties (Table S1), these records exhibit a consistent pattern that suggests that the North Atlantic SPG weakened twice. Such a weakening of the gyre has been observed in modelling studies in response to a sea-ice driven dampening of subpolar North Atlantic convection^{4,7,8}. The evidence presented here for loss of resilience in the subpolar North Atlantic before 1260, together with evidence for the weakening^{18,44} of the potentially bistable^{10,11,14} subpolar gyre, indicate that the onset of the LIA may have occurred in response to the subpolar gyre system passing a tipping point.”

4. Similarly, the first period also appears to have been one of increased sea-ice in the subpolar gyre (Labrador Sea) due to the “warmer” MCA period triggering calving and sea ice export (lines 171-174). One question raised: Are these two episodes of “declining resilience” caused by warming SST? It is not explicitly put forward in the current manuscript, but it seems to be plausible.

Thank you for this observation. We now include a substantial discussion around potential drivers, which incorporates the interpretation raised by the reviewer (and put forward in Lapointe and Bradley, 2021) within the discussion.

We have added the following sentence to page 9, lines 221-225: “As for the second episode, it is possible that the cooling effect caused by the strong volcanic eruptions⁵⁷ and the Wolf solar minimum during the second half of the 13th century (Fig. 4k,l) had contributed to the expansion of Arctic sea-ice³¹, which was later exported into subpolar latitudes and its melting potentially enhanced by the warm episode during the 14th century⁴³.”

5. The “weakening of negative feedbacks” may be linked with the accumulation of arctic sea ice in the arctic high latitudes during the late 1300s (Miles et al. 2020). This, in turn, may be consistent with persistent atmospheric blocking over Greenland in times

of warmer SST in the subpolar gyre as shown by many proxies in the Labrador Sea, as was the case in the 1960s (Ionita et al. 2016, Lapointe and Bradley 2021).

Thank you for suggesting this potential linkage between the position of the center of atmospheric blocking and sea ice export through Fram Strait. We have mentioned the mechanism proposed by Ionita et al. (2016) in the discussion about the potential role of external forcing.

We have added the following sentence to page 9, lines 213-215: “This condition has also been suggested to strengthen the northeasterly winds, flushing sea ice from the Arctic into the subpolar North Atlantic⁵³, reinforcing the weakening of the SPG.”

6. As for the interpretation of the second “destabilization” period: the two peaks in very low values of $\delta^{18}\text{O}$ at the end of the 1300s is in phase with the significant $\text{AR}(1)$ ($\tau = 0.90$, $p < 0.001$). Wouldn't that be expected that autocorrelation in the $\delta^{18}\text{O}$ data be large during this period of increasingly lower values (Reynolds et al. 2016)? I would like the authors comment on that.

Trends towards extreme values in a record can generate increasing autocorrelation, but we detrended the records before calculating $\text{AR}(1)$ and variance on the residuals, so the signal is related to the fluctuations not the trend. The robustness analysis shows that overall, the positive trend in $\text{AR}(1)$ is maintained after using short detrending bandwidths (Fig. 3), which are expected to attenuate the irregular oscillations. To further confirm the results from the robustness analysis we have included a figure in the supplementary material (Fig. S2) that shows the residuals of the records after removing the trend using a bandwidth of 12 years. In the residuals, there is no trace of the increasingly lower values associated with the incursion of warm Atlantic waters during the second episode (Fig. S2c) indicating that it is the fluctuations that exhibit intrinsic critical slowing down. Thank you for this observation, we have addressed that possibility on the main text (pages 5-6, lines 128-131).

Minor comments:

7. Figure 1: indicate all the letters (sites name) in the caption (a to e).

Thank you, the suggested change has been implemented (Page 19, lines 344-347).

8. Line 64: three annually resolved proxy records. These proxies are $\delta^{18}\text{O}$, $\delta^{13}\text{C}$ and band width from the same location. Am I right?

Thanks for pointing out this lack of clarity, we have rephrased the sentence to refer to the three bivalve-derived proxy records (Page 3, line 67).

9. Line 99: Do you imply that 1400 was the onset of the Little Ice Age? If so, it would be probably good to define it in the introduction.

In terms of the stability, we would assert that after ~1400 the Subpolar North Atlantic is in its Little Ice Age state. This is a sensible point to pull out, thank you. We have rephrased that sentence (page 4 lines 100-101) and mention in the introduction that the first signs of the LIA cooling are reported in the Arctic and Northern Europe during the 13th century.

10. Lines 181-184: The argument of a rapid drop in SST following the first and especially the second destabilization events is not very convincing for the summer SST reconstruction based on alkenone palaeothermometry (Fig. 4b), i.e., the SST drop for this proxy occurs very much in phase with the second destabilization event. Also, the drop of SST in the Vøring Plateau is not too obvious after the first episode.

We agree. In our latest compilation of reconstructions, we decided to remove the alkenone-based proxy for SST in the North Icelandic Shelf as this record does not reproduce the recent observed warming trend. Instead, we included a diatom-based proxy record for SST from the same location (Fig. 4f). In this new record and the one from the Vøring Plateau, the first cooling episode is not as evident as in other records. Also, it is possible that the observed cooling around 1300 in North Iceland was associated with the presence of sea ice (Fig. 4a) masking the effect of the potential weakening of the subpolar gyre around 1400. We want to highlight that we included the records that met our criteria even if not all of them exhibit clear rapid changes after each episode. While the SST picture might not be very clear, we find overwhelming evidence from high-resolution well-dated records that a circulation weakening followed each destabilisation episode (see response to Reviewer 2, question 3).

11. Lines 196-197: the initial cooling may have been caused by warming conditions during the MCA triggering calving and sea ice export. It would be needed to also discuss the 1257 Samalas eruption and the possible positive (sea ice) feedbacks associated with it.

Thank you for this observation. Following another referee's suggestion, we have included in the discussion the potential role of external forcing driving each destabilisation episode (page 9, lines 207-225). Within that paragraph, we have discussed the potential role of the sequence of volcanic eruptions around the MCA-LIA transition. In summary, we find no evidence linking the first destabilization episode with external forcing because it occurred before land records showed signs of the initial cooling that may have triggered Arctic sea ice expansion and its subsequent export into subpolar latitudes. In contrast, the timing of the second episode may be consistent with the hypothesis of reduced radiative forcing triggering sea ice expansion.

References included in the response

- Alonso-Garcia, M., Kleiven, H. (Kikki) F., McManus, J. F., Moffa-Sanchez, P., Broecker, W. S., & Flower, B. P. (2017). Freshening of the Labrador Sea as a trigger for Little Ice Age development. *Climate of the Past*, 13(4), 317–331.
- Andresen, C., Hansen, M., Seidenkrantz, M.-S., Jennings, A., Knudsen, M., Nørgaard-Pedersen, N., Larsen, N., Kuijpers, A., & Pearce, C. (2013). Mid- to late-Holocene oceanographic variability on the Southeast Greenland shelf. *The Holocene*, 23(2), 167–178.
- Andrews, J. T., & Jennings, A. E. (2014). Multidecadal to millennial marine climate oscillations across the Denmark Strait (~ 66° N) over the last 2000 cal yr BP. *Climate of the Past*, 10(1), 325–343.
- Ionita, M., Scholz, P., Lohmann, G., Dima, M., & Prange, M. (2016). Linkages between atmospheric blocking, sea ice export through Fram Strait and the Atlantic Meridional Overturning Circulation. *Scientific Reports*, 6(1), 32881.
- Lapointe, F., & Bradley, R. S. (2021). Little Ice Age abruptly triggered by intrusion of Atlantic waters into the Nordic Seas. *Science Advances*, 7(51), eabi8230.
- Miettinen, A., Divine, D. V., Husum, K., Koç, N., & Jennings, A. (2015). Exceptional ocean surface conditions on the SE Greenland shelf during the Medieval Climate Anomaly. *Paleoceanography*, 30(12), 1657–1674.
- Moffa-Sánchez, P., & Hall, I. R. (2017). North Atlantic variability and its links to European climate over the last 3000 years. *Nature Communications*, 8(1), 1726.
- Moffa-Sanchez, P., Hall, I. R., Thornalley, D. J. R., Barker, S., & Stewart, C. (2015). Changes in the strength of the Nordic Seas Overflows over the past 3000 years. *Quaternary Science Reviews*, 123, 134–143.

Reviewers' Comments:

Reviewer #1:

Remarks to the Author:

The authors have satisfactorily addressed the needed points from this reviewer. They have now addressed the potential drivers behind the first destabilization episode. They also now discuss the possible role of external forcing from volcanism and solar variability. Both aspects are treated well and satisfying, without bringing in uncertainties to the authors' main findings. The authors have also made a more distinction between interpreting sea ice vis-à-vis calving of glacial ice.

The authors have also made an engaged effort to address the other reviewer's insightful comments, notably the justification of the paleo records included and several others not included in the original submission. As a result, the overall reconstruction of events is enhanced and is treated with appropriate additional discussion, again without bringing in uncertainties to the authors' main findings.

I commend the authors and recommend the manuscript be published in its present form.

Reviewer #2:

Remarks to the Author:

The authors have now included a thorough discussion addressing the questions of the reviews. Also, the addition of several marine records brings more credibility and improve the overall interpretation. This paper will be of great interest for the community and I am happy to recommend this paper in Nature Communications.

Minor comments:

Figs. 2 and 4: add a tick each 100 years

Typo at Line 168: "Sa"

Response to Reviewers

We would like to thank again the reviewers for their considered and valuable comments which have helped to enhance the interpretation of our results. We are pleased they have found our findings interesting for the community and recommended its acceptance.

Reviewer #1 (Remarks to the Author):

The authors have satisfactorily addressed the needed points from this reviewer. They have now addressed the potential drivers behind the first destabilization episode. They also now discuss the possible role of external forcing from volcanism and solar variability. Both aspects are treated well and satisfying, without bringing in uncertainties to the authors' main findings. The authors have also made a more distinction between interpreting sea ice vis-à-vis calving of glacial ice.

The authors have also made an engaged effort to address the other reviewer's insightful comments, notably the justification of the paleo records included and several others not included in the original submission. As a result, the overall reconstruction of events is enhanced and is treated with appropriate additional discussion, again without bringing in uncertainties to the authors' main findings.

I commend the authors and recommend the manuscript be published in its present form.

Reviewer #2 (Remarks to the Author):

The authors have now included a thorough discussion addressing the questions of the reviews. Also, the addition of several marine records brings more credibility and improve the overall interpretation. This paper will be of great interest for the community and I am happy to recommend this paper in Nature Communications.

Minor comments:

Figs. 2 and 4: add a tick each 100 years

Thank you for suggesting this visual improvement, we have now added 100-yr ticks to both figures.

Typo at Line 168: "Sa"

We have changed it to "S3a".